# Integration of miRNA:mRNA Co-Expression Revealed Crucial Mechanisms Modulated in Immunogenic Cancer Cell Death

**DOI:** 10.3390/biomedicines10081896

**Published:** 2022-08-05

**Authors:** María Julia Lamberti, Barbara Montico, Maria Ravo, Annunziata Nigro, Giorgio Giurato, Roberta Iorio, Roberta Tarallo, Alessandro Weisz, Cristiana Stellato, Agostino Steffan, Riccardo Dolcetti, Vincenzo Casolaro, Damiana Antonia Faè, Jessica Dal Col

**Affiliations:** 1Department of Medicine, Surgery and Dentistry “Scuola Medica Salernitana”, University of Salerno, 84081 Baronissi, SA, Italy; 2INBIAS, CONICET-UNRC, Río Cuarto, Córdoba 5800, Argentina; 3Immunopathology and Cancer Biomarkers, Centro di Riferimento Oncologico di Aviano (CRO), IRCCS, 33081 Aviano, PN, Italy; 4Genomix Life Srl, 84081 Baronissi, SA, Italy; 5Laboratory of Molecular Medicine and Genomics, Department of Medicine, Surgery and Dentistry “Scuola Medica Salernitana”, University of Salerno, 84081 Baronissi, SA, Italy; 6Peter MacCallum Cancer Centre, Melbourne, VIC 3000, Australia; 7Sir Peter MacCallum Department of Oncology, The University of Melbourne, Melbourne, VIC 3010, Australia; 8Department of Microbiology and Immunology, The University of Melbourne, Melbourne, VIC 3010, Australia

**Keywords:** immunogenic cell death, miRNA, MHC-II, antigen presentation pathway

## Abstract

Immunogenic cell death (ICD) in cancer represents a functionally unique therapeutic response that can induce tumor-targeting immune responses. ICD is characterized by the exposure and release of numerous damage-associated molecular patterns (DAMPs), which confer adjuvanticity to dying cancer cells. The spatiotemporally defined emission of DAMPs during ICD has been well described, whereas the epigenetic mechanisms that regulate ICD hallmarks have not yet been deeply elucidated. Here, we aimed to examine the involvement of miRNAs and their putative targets using well-established in vitro models of ICD. To this end, B cell lymphoma (Mino) and breast cancer (MDA-MB-231) cell lines were exposed to two different ICD inducers, the combination of retinoic acid (RA) and interferon-alpha (IFN-α) and doxorubicin, and to non ICD inducers such as gamma irradiation. Then, miRNA and mRNA profiles were studied by next generation sequencing. Co-expression analysis identified 16 miRNAs differentially modulated in cells undergoing ICD. Integrated miRNA-mRNA functional analysis revealed candidate miRNAs, mRNAs, and modulated pathways associated with Immune System Process (GO Term). Specifically, ICD induced a distinctive transcriptional signature hallmarked by regulation of antigen presentation, a crucial step for proper activation of immune system antitumor response. Interestingly, the major histocompatibility complex class I (MHC-I) pathway was upregulated whereas class II (MHC-II) was downregulated. Analysis of MHC-II associated transcripts and HLA-DR surface expression confirmed inhibition of this pathway by ICD on lymphoma cells. miR-4284 and miR-212-3p were the strongest miRNAs upregulated by ICD associated with this event and miR-212-3p overexpression was able to downregulate surface expression of HLA-DR. It is well known that MHC-II expression on tumor cells facilitates the recruitment of CD4+ T cells. However, the interaction between tumor MHC-II and inhibitory coreceptors on tumor-associated lymphocytes could provide an immunosuppressive signal that directly represses effector cytotoxic activity. In this context, MHC-II downregulation by ICD could enhance antitumor immunity. Overall, we found that the miRNA profile was significantly altered during ICD. Several miRNAs are predicted to be involved in the regulation of MHC-I and II pathways, whose implication in ICD is demonstrated herein for the first time, which could eventually modulate tumor recognition and attack by the immune system.

## 1. Introduction

Cancer immunoediting theory defines a three-step process that describe the impact of the immune system on shaping tumor development. Those steps are called elimination, equilibrium, and escape [1,2]. During the first two phases, cancer cells are eliminated by the immune system, a process called immunosurveillance, which results in the prevention of tumor growth. Under the persistent selective pressure by the immune system, cancer cells acquire genetic and epigenetic alterations that permit them to grow even with a concomitant anti-tumor immune response. Resistant clones selected from the equilibrium phase escape immune control through multiple mechanisms to enable tumor progression and metastasis [3]. The regulation of cancer immunoediting is dynamic, heterogeneous, and context-dependent. In this respect, microRNAs (miRNAs) have been well characterized as important regulators of both innate and adaptive antitumor immunity, participating in both protection against cancer development and shaping the hallmarks of emerging tumor cells able to avoid immunosurveillance [4,5,6]. miRNAs are short non-coding RNAs that regulate gene expression through translation blockade or mRNA degradation [7].

Several immunotherapeutic approaches can reinstate immunosurveillance to provide long-term survival of cancer patients. The beneficial outcome that is offered by certain agents for cytotoxic chemotherapy (doxorubicin, mitoxantrone, shikonin, colchicine) or physical therapies (high hydrostatic pressure, photodynamic therapy, radiotherapy, UV-light, hyperthermia) depends on their ability to induce immunogenic cell death (ICD), which positively contributes to immune-mediated recognition of tumors [8]. Regardless of the stimuli, ICD triggers a cascade of events that begins with a premortem stress that concomitantly allows the spatiotemporally coordinated exposure and/or release of immunomodulatory molecules (damage-associated molecular patterns, DAMPs) and tumor cell contents. DAMPs released during ICD are critical for successful tumor-antigen presentation by activating antigen presenting cells (APCs), and the development of tumor-specific adaptive immune responses. Thus, ICD makes dying cancer cells immunogenic by enhancing both antigenicity and adjuvanticity [9,10,11,12]. 

In recent years, knowledge about the identity of ICD-associated DAMPs as well as their modulation, mainly at post-translational level [13,14,15], has increased remarkably. However, transcriptional changes induced by ICD are not fully elucidated, nor are the genetic and epigenetic pathways involved in this phenomenon. Recently, Rufo et al. explored the transcriptional inflammatory signature that distinguishes immunogenic from not-immunogenic therapies [16]. Through their study, they discovered “multi-gene expression patterns” or “metagenes” as prognostic biomarkers and describe post-transcriptional or post-translational modifications that play an important role in governing the immunological signaling outcome [16]. In this scenario, we recently recapitulated the data available about the effect of miRNAs on DAMPs expression [17]. Surprisingly, we noticed that, at present, miRNA modulation during ICD and its contribution to this process is still largely underexplored.

Based on these considerations, we proposed here to define miRNAs expression pattern of cancer cells undergoing ICD and to identify the potential contribution of these small non-coding sequences to cancer cell immunogenicity. To this end, we adopted our previously described model of immunogenic apoptosis in which ICD is induced in lymphoma cell lines by the combination of retinoic acid (RA) and interferon-α (IFN-α) [18,19,20]. We have demonstrated that the RA/IFN-α combination induced immunogenic apoptosis characterized by expression of ICD hallmarks. Consistently, RA/IFN-α-treated apoptotic cells were preferentially recognized and engulfed in vitro by monocyte-derived dendritic cells (DCs) compared to controls. In addition, previous results indicated that DCs loaded with immunogenic tumor cell lysates obtained from RA/IFN-α-treated lymphoma cell lines were highly efficient in eliciting tumor and antigen-specific cytotoxic T lymphocytes in vitro and in mediating tumor growth inhibition in vivo [18,19,20].

In this study, we performed next generation sequencing (NGS)-based RNA sequencing to show that ICD inducers were capable of significantly modifying the miRNA signature of dying tumor cells. Inverse co-expression analysis defined clusters of miRNA:mRNA pairs modulated during ICD. In silico functional profiling of highly predictive target genes revealed their involvement in immune function, in particular antigen presentation pathways. Further validation experiments confirmed for the first time the regulation of class I and II MHC by ICD inducers suggesting the involvement of these molecular pathways in promoting tumor recognition and control by innate and adaptive immune cells. Moreover, our results indicate the potential involvement of miR-27a-5p and miR-222-5p in ICD-related class I MHC upregulation and of miR4284 and miR-212-3p in class II MHC downregulation. 

## 2. Materials and Methods

### 2.1. Cell Lines and Culture Conditions

Mino (human mantle cell lymphoma, ATCC CRL-3000), DOHH2 (human diffuse large B-cell lymphoma, DSMZ ACC 47), and Meljuso (melanoma, DSMZ ACC 74) cell lines were cultured in RPMI 1640 (Sigma, now Merck, Darmstadt, Germany), while MDA-MB-231 cell line (human breast adenocarcinoma, ATCC HTB-26) was cultured in DMEM (Sigma, now Merck, Darmstadt, Germany); both media were supplemented with 10% FCS (Fetal Calf Serum, Gibco, Paisley, Scotland), 100 μg/mL streptomycin, and 100 IU/mL penicillin (Sigma, now Merk, Darmstadt, Germany) and maintained at 37 °C in a humidified 5% CO_2_ atmosphere. 

### 2.2. Reagents

Cell death was induced with human IFN-α (IntronA SP Europe) and 9-cis-Retinoic Acid (Sigma, now Merck, Darmstadt, Germany) used at 1000 U/mL and 1 µmol/L, respectively, for 48 h, or Doxorubicin (Sigma, now Merck, Darmstadt, Germany) used at 0.5 µM, for 24 h. Control tumor cells were irradiated with 80 gray for 16 min [20].

### 2.3. RNA Purification, Sequencing, and Data Analysis

Total RNA was extracted as described previously [21]. For Total RNA-Sequencing, indexed libraries were prepared from 1 µg/ea. purified RNA with Tru-Seq Stranded Total RNA Sample Prep Kit (Illumina Inc., San Diego, CA, USA) according to the manufacturer’s instructions. Libraries were sequenced (paired-end, 2 × 75 cycles) on NextSeq500 platform (Illumina Inc. San Diego, CA, USA). The raw sequence files generated (.fastq files) underwent quality control analysis using FastQC (http://www.bioinformatics.babraham.ac.uk/projects/fastqc/, accessed on 30 May 2022) and the quality checked reads were then aligned to the human genome (hg19 assembly) using TopHat version 2.0.10 [22], with standard parameters. A given mRNA was considered expressed when detected by at least 10 reads. Differentially expressed mRNAs were identified using DESeq2 [23]. Firstly, gene annotation was obtained for all known genes in the human genome, as provided by Ensemble (GRCh37). Using the reads mapped to the genome, we calculated the number of reads mapping to each transcript with HTSeq-count [24]. These raw read counts were then used as input to DESeq2 for calculation of normalized signal for each transcript in the samples, and differential expression was reported as Fold Change along with associated FDR. For Small RNA sequencing, indexed libraries were prepared from 1 µg/ea. purified RNA with TruSeq small RNA Sample Prep Kit (Illumina Inc., San Diego, CA, USA) according to the manufacturer’s instructions. Sized smallRNA libraries were gel purified and sequenced on NextSeq500 (Illumina Inc., San Diego, CA, USA). smallRNA-Seq data were analyzed using iSmart [25] setting the default parameters. A given miRNA was considered expressed when detected by at least 3 reads, while differentially expressed miRNAs were identified using DESeq2.

### 2.4. UpSet Plot 

Since a Venn diagram to present relationships between interactive sets would be complicated when dealing more than three sets, UpSet plots [26] were used to analyze the intersections between the six combinations of cell line and treatment condition. UpSet plots were created in R using the UpSetR library [27].

### 2.5. RNA Extraction, cDNA Synthesis, and Quantitative Real-Time PCR (RT-qPCR)

According to the manufacturer’s recommendations, an appropriate number of cells was collected and washed in PBS. Total RNA was extracted with QIAGEN miRNeasy Mini Kit (Qiagen, Venlo, The Netherlands), eluted in 30 µL and quantified using the Nanodrop 2000c (Thermo Scientific, Wilmington, DE, USA).

For Real-Time PCR analysis of transcripts 1 µg of RNA was retro-transcribed with LunaScript RT SuperMix (NED), following manufacturer’s instruction. Real-time PCR was performed using FluoCycle II™ SYBR^®^ Master Mix (Euroclone SpA, Milan, Italy). Each reaction of RT-qPCR contains 10 µL of FluoCycle™ II SYBR^®^ Master Mix, 1.2 µL of forward and reverse primers (10 µM), 50 ng of cDNA, and high-quality distilled water in a final volume of 20 µL. Primers were designed with PrimerQuest™ Tool (IDT) and were provided by IDT.

Primer sequences were the following: 

HLA-DQB1: forward CTCTTCTGTGATGCCTGCTTAT, reverse GCCACTGTAGGACTTTGATCTC; HLA-DRB5: forward TACCTAACCCTACGGCCTCC, reverse CTTTGGAGCCAGGTGGACAG; HLA-DOA: forward TTTGCCCGCTTTGACCCGCA, reverse TCACCCGTGGAGGCACGTTG; CIITA: forward CTGTGCCTCTACCACTTCTATG, reverse GTCGCAGTTGATGGTGTCT; HLA-DRA: forward GACGATTTGCCAGCTTTGAG, reverse GGTACATTGGTGATCGGAGTATAG; CD74: forward CCCAAGGAAGAGCCAATGT, reverse CATGGCCCTGAAAGCTGATA; GAPDH: forward TGCACCACCAACTGCTTAGC, reverse GGCATGGACTGTGGTCATGAG.

For miRNA quantification in RT-qPCR, 500 ng of total RNA were retro-transcribed using Applied Biosystems^®^ TaqMan^®^ MicroRNA Reverse Transcription Kit and specific primers for each microRNA of interest. Then, RT-qPCR was performed through specific probes using Applied Biosystems^®^ TaqMan™ Universal PCR Master Mix. Each reaction of RT-qPCR contains 10 µL of Master Mix, 1 µL of TaqMan^®^ miRNA Assay (20X), 50 ng of cDNA, and high-quality distilled water in a final volume of 20 µL.

Specific amplicons were quantified by RT-qPCR using the LightCycler ^®^480 Real-Time PCR System (Roche, Basel, Switzerland); cycling condition consisted in one step at 95 °C for 10 min, two steps 95 °C for 15 s/60 °C for 1 min repeated 45 times, a curve melting cycle (for transcripts), from 65 °C to 95 °C, with an increment of 0.6 °C. Normalized relative quantity was calculated with “ΔΔ-Ct” method, using U6 and GAPDH as housekeeping for miRNA and mRNA, respectively.

### 2.6. Performance and Visualization of Enrichment Analysis of Differentially Expressed Genes

To examine the biological functions of differentially expressed genes by ICD (Fold Change FC ≥ 1.3; *p*-value < 0.05), enrichment analyses were performed based on Gene Ontology (GO) and Reactome Pathway databases, using g:GOST tool in gProfiler (https://biit.cs.ut.ee/gprofiler/ accessed on 16 November 2021) [28]. The Ggplot2 package in R was used for bar and bubble plotting [29].

### 2.7. Visualization and Analysis of miRNA—Targets Interactions

miRNA target prediction was performed using miRWalk v.3.0 (http://mirwalk.umm.uni-heidelberg.de/ accessed on 15 July 2021) [30], which calculates a score for each putative miRNA–mRNA interaction. As predicted miRNA targets, we considered mRNAs characterized by a prediction score ≥0.9 and searching the complete transcript sequence including the 5′-UTR, CDS, and 3′-UTR. Differentially expressed mRNAs and miRNA with FC ≥ 1.3 and *p*-value < 0.05 were considered for the analysis. To visualize and analyze the regulatory networks composed with inverse co-expressed miRNA::mRNA predicted target pairs, we used Cytoscape, as previously described [31]. Since networks generated by the miRNA::mRNA predictions had a large number of miRNA–mRNA interactions, we focused only on those whose genes were associated to GO Term: Immune System Process (GO:0002376, data obtained from g:Convert tool in gProfiler). To identify clusters in the complex networks generated, ClusterViz and the EAGLE algorithm with default options was used [32]. This analysis results in major sub-networks. Here, the first clusters, with the largest number of nodes (miRNAs and genes) and edges (interactions) and a significant degree of modularity (~0.5 [11]), were selected for additional analysis. Next, BINGO v.3.0.3 was employed to calculate overrepresentation of Gene Ontology (GO) “biological process” terms among the input gene lists, corresponding to cluster 1. Functional enrichment visualization was created by the enrichment map plugin in Cytoscape using the exported BINGO results as the input [33]. 

### 2.8. Gene Expression Analysis from GEO Database

A dataset containing RNA-seq data about gene expression modulation by ICD was downloaded from Gene Expression Omnibus database (GEO ID: GSE163377). Briefly, the human melanoma A375 cell line was subjected to treatment with ICD inducers such as mitoxantrone (MTX) or hypericin-based photodynamic therapy (Hyp-PDT) and to non-ICD inducer cisplatin (CDDP). Samples for RNA-seq were collected at 0, 4, 10, and 20 h [16]. Expression values in this dataset were normalized to control at 0 h.

### 2.9. Flow Cytometry

Surface staining of single-cell suspensions of tumor cells was performed as previously described [18] and analyzed on a BD FACSVerse™ Flow Cytometer (2-laser, 6-color (4-2) configuration. An HLA-DR-FITC antibody (980402, BioLegend, San Diego, CA, USA) was used to analyze MHC-II expression and 7-AAD (Miltenyi Biotec, Bergisch Gladbach, Germany) to exclude dead cells. Data analysis was conducted using FlowJo X software (version 10.0.7, BD, Ashland, OR, USA).

### 2.10. Oligonucleotides Transfection

Synthetic hsa-miR-4284 and hsa-miR-212-3p mimics (miRCURY LNA miRNA Mimics) or the appropriate scrambled controls (negative control, NC) were purchased from Qiagen. Oligonucleotides and were transfected (75 nM) in Meljuso cell line using TransIT-siQUEST Transfection Reagent (Mirus Bio LLC, Madison, WI, USA), according to the manufacturers’ recommendations. After 48 h, the cells were harvested and the total RNA was extracted.

### 2.11. Data Availability

Raw RNA sequencing data and smallRNA-Seq data are deposited in the EBI ArrayExpress database (http://www.ebi.ac.uk/arrayexpress) with accession number E-MTAB-11799 and E-MTAB-11803 respectively (accessed on 7 April 2022).

### 2.12. Statistics

Data handling, analysis, and graphic representation (all shown as mean ± SEM) were performed using Prism v7.0 (GraphPad Software, San Diego, CA, USA). Statistical significance was calculated by Dunnett’s multiple comparisons test to one- or two-way ANOVA analysis, and a *p* value < 0.05 was considered statistically significant (* *p* < 0.05, ** *p* < 0.01, *** *p* < 0.001).

## 3. Results

### 3.1. miRNA Expression Signature in Immunogenic Cancer Cell Death

To comprehensively define the miRNA profile associated to ICD, we performed RNA sequencing (RNA-seq) of lymphoma (Mino) and breast cancer (MDA-MB-231) cell lines responding to the combination RA/IFNα or doxorubicin (DXR), as prototypes of immunogenic treatments, whereas gamma-irradiation (gIRR) served as a model non-ICD inducer [20]. To visualize the expression overlap between regulated miRNAs in each condition, UpSet plots were drawn. Our findings showed a signature of 16 miRNAs differentially modulated by both ICD inducers in lymphoma and breast cancer cell lines, whose expression was not affected by non-ICD inducer (Table 1). Among them, 9 miRNAs were downregulated (Figure 1a) whereas 7 were upregulated (Figure 1b) in cells undergoing ICD. Next, we confirmed by RT-qPCR that miR-4284 and miR-212-3p were significantly upregulated in both cancer models in response to at least one of the used ICD inducers, whereas miR-7705 was significantly upregulated only in MDA-MB-231 following RA/IFNα treatment (Figure 1c,d). Moreover, the combination of RA/IFNα increased miR-4284 in an additional lymphoma cell line (DOHH2) (Appendix A) in which we previously documented ICD induction by this treatment [20].

### 3.2. mRNA Expression Signature in Immunogenic Cancer Cell Death

We then interrogated the ICD-dependent changes in the transcriptome of the treated lymphoma and breast cancer cells by RNA-seq analysis. In concordance to miRNAs analysis, we specified this core profile as genes that were regulated by RA/IFNα and DXR, but not by gIRR, in both Mino and MDA-MB-231 cell lines (Table 2). Of note, the expression of 6 genes was downregulated by ICD inducers (Figure 2a), and 35 genes exhibited enhanced expression (Figure 2b). Enrichment analysis performed on the set of significantly upregulated genes determined “Immune System”, “Class I MHC mediated antigen processing and presentation”, and “Cytokine Signaling in system” as the most significant Reactome pathways associated with immunogenic treatments (Figure 2c). Importantly, antigen presentation-related GO terms were dramatically enriched among the upregulated genes, including those involved in molecular functions (e.g., “MHC class I protein binding”), cellular components (e.g., “TAP complex”), and biological processes (e.g., “antigen processing and presentation antigen via MHC class I”) (Figure 2d). Together, these data suggest that ICD induced a distinct transcriptional signature hallmark by regulation of antigen presentation, a crucial step for proper activation of the immune system antitumor response.

### 3.3. Identification of Immunogenic miRNA–mRNA Pairs by Inverse Co-Expression Networks of Immune System-Associated Genes

We have previously described a model of immunogenic apoptosis in which ICD is induced in lymphoma cell lines by RA/IFN-α, characterized by enhanced calreticulin, Hsp70 and Hsp90 surface exposure, and HMGB1 secretion [20]. To further unravel the potential role of miRNAs in the transcriptional signature modulated by ICD, we performed a subsequent in silico inverse co-expression analysis of ICD-associated-miRNAs (Table 1) with RNA-seq data from RA/IFNα-treated Mino cells. To identify putative target genes regulated by miRNAs, we employed the computational prediction tool miRWalk. Briefly, 1710 predicted interactions were found between the 7 upregulated ICD-miRNAs and the 578 mRNAs downregulated by RA/IFNα, and 3228 predicted interactions between the 9 downregulated ICD-miRNAs and the 1089 mRNAs upregulated by RA/IFNα (Figure 3a,c). From these data, immunogenic miRNA-mRNA target pairs were defined as the pairs of both elements which: (a) showed opposite expression patterns; (b) exhibited sequence-based target relationship predicted by miRWalk; and (c) included a gene connected to the GO Term: Immune System Process. We identified 313 and 901 immunogenic miRNA–mRNA target pairs, for up- and downregulated ICD-miRNAs, respectively (Figure 3a,c). Next, using Eagle algorithm via ClustalViz, the networks were subdivided into 4 sub-clusters for upregulated ICD-miRNAs (Figure 3b) and 5 sub-clusters for downregulated ICD-miRNAs (Figure 3d). Each sub-cluster is composed by one or more miRNA (triangles) surrounded by genes (circles) whose expression was predicted to be regulated by the miRNA in the center. The first clusters, with the largest number of nodes (miRNAs and genes) and edges (interactions) and a significant degree of modularity (~0.5 [34]), were selected for further analysis. Among upregulated ICD-miRNAs, the first cluster included miR-4284 and miR-212-3p (Table 3), whereas for downregulated ICD-miRNAs, the first cluster included hsa-miR-27a-5p and hsa-miR-222-5p (Table 4). 

### 3.4. Enrichment Analysis of Target Genes of miRNA Members in Most Representative Clusters

To understand the biological and functional implications of these miRNAs and their putative targets in ICD, BiNGO enrichment analysis was performed for each cluster independently using Cytoscape. The results of BiNGO analysis showed that variations in upregulated genes linked to biological processes were mainly enriched in “immune system process”, in particular “antigen presentation pathway via MHC-I” (Figure 4a). On the other hand, in this context, enrichment analysis demonstrated that the downregulated genes in Cluster 1 were involved in “antigen presentation pathway via MHC-II” (Figure 4b). Modulation of Class I and Class II MHC transcripts by ICD and not-ICD regimens was also explored using a meta-analysis approach based on data published on GEO database (GEO ID: GSE163377). These gene sets include the RNA-seq data of A375 melanoma cell responding to mitoxantrone (MTX) or hypericin-based photodynamic therapy (Hyp-PDT), as immunogenic treatments, whereas cisplatin (CDDP) served as the experimental model of non-ICD inducer [16]. Regarding antigen presentation associated-transcripts, immunogenic treatments elicited similar kinetics of downregulation of Class II genes, whereas they upregulated (MTX) or did not affect (Hyp-PDT) Class I transcripts expression. Non-immunogenic treatment (CDDP) did not induce significant alterations on antigen presentation pathways (Figure 5). Overall, this evidence suggests that downregulation of Class II MHC could be proposed as a novel feature of ICD.

### 3.5. Validation of ICD-Mediated Decrease of Class II MHC Antigen Presentation Pathway via Upregulation of miR-212-3p

To confirm the effect of ICD inducers on MHC-II Antigen Presentation Pathway, its regulation was further explored in lymphoma cell lines (Mino and DOHH2) exposed to RA/IFNα and DXR. Since miRNAs modulate many of their targets through mRNA transcript degradation or translation inhibition [35], we determined the expression of MHC-II-associated molecules at both protein and mRNA levels. Lymphoma cells subjected to ICD were analyzed by flow cytometry to assess the membrane expression of HLA-DR. Surface HLA-DR was decreased in both RA/IFNα-treated lymphoma cell lines, whereas DXR downregulated its expression only in Mino cells (Figure 6a,b). The RT-qPCR results showed that, as expected from the RNA-Seq data, expression of most MHC-II associated transcripts (CD74, HLA-DOA, HLA-DQB1, HLA-DRA, CIITA) was downregulated in Mino cells by RA/IFNα (Figure 6c) and DXR (Appendix A) treatments whereas no significant changes were detected in the DOHH2 cell line (Figure 5c), suggesting that miRNAs could mediate gene regulation in a cell type-specific manner. Accordingly with our hypothesis, the not-ICD inducer γ-irradiation did not regulate MHC Class II pathway (Appendix A). Next, we analyzed in the same samples the expression of miR-4284 and miR-212-3p. miR-4284 upregulation was confirmed in both lymphoma cell lines following RA/IFNα treatment, whereas the expression of miR-212-3p was only induced in Mino cells (Figure 7a). We then evaluated the direct contribution of the selected miRNAs to MHC-II downregulation through the employment of miRNA mimics. However, Mino cells, having a B-cell phenotype, physiologically express MHC-II molecules and the Toll-like receptors (TLR) 3, 7, and 8. TLR3, TLR7, and TLR8 are responsible for recognition/binding of single and double stranded RNA and for consequent triggering of a signal cascade resulting in MHC-II upregulation as off-target effect [36]. Therefore, the transfection of Mino with specific miRNA mimics as well as negative control RNAs increased the expression of different transcripts of the MHC-II pathway to a similar extent (data not shown). To overcome this problem, the Meljuso melanoma cell line was used for transfection experiments as proposed by van Paul et al. [37]. In fact, Meljuso cells express the MHC-II complex but do not express TLRs. Meljuso cells were transfected with miR-212-3p and miR-4284 mimics, their combination, or negative control (NC) for 48 h. In flow cytometry, cells transfected with miR-212-3p or both mimics displayed a slow but significant and consistent surface HLA-DR downregulation compared to the NC-transfected cells (Figure 7b,c). The RT-qPCR results showed that expression of HLA-DRA, CIITA, and HLA-DOA was downregulated in the sample where the mimics were combined, or, in the case of CIITA and HLA-DOA, by individual miRNA mimics (Figure 7d). Altogether, these results demonstrated that miRNAs upregulated by ICD modulate the expression of MHC class II molecules. In particular, miR-212-3p exerted a significant inhibition of the MHC-II pathway at both posttranscriptional and translational levels. 

## 4. Discussion

ICD offers a new opportunity to expand the success of cancer treatment, due to the combination of killing tumor cells with acting as a “tumor vaccine”, triggering a tumor-specific immune response against live cancer cells and residual tumor tissue. In this way, patients can achieve long-term clinical benefits favored by immunogenic chemo- or physical therapies endowed with intrinsic adjuvanticity [9,10,11,12].

The capacity of ICD to drive adaptive immunity relies on the enhancement of two major parameters: antigenicity and adjuvanticity. Antigenicity depends on the expression and presentation of tumor-specific antigens that could be recognized by naïve T cell clones. Adjuvanticity is supported by the spatiotemporally coordinated release or exposure of DAMPs that are required for the enrolment and immunogenic activation of antigen-presenting cells (APCs) [12]. 

DAMPs modulated by ICD must be exposed at sufficient levels to promote a robust DC stimulation. The kinetics and intensity of their release are regulated by intracellular responses triggered by the initiating stressor [9,13,38,39]. In this respect, DAMPs have been classified as constitutive or inducible. Briefly, constitutive DAMPs (cDAMPs) are endogenous molecules that are constitutively expressed before death and are released/exposed by dying cells (e.g., calreticulin, HMGB1, ATP), whereas inducible DAMPs (iDAMPs) are the endogenous molecules generated via mechanisms that are dependent on the underlying cell death pathway activated (e.g., type 1 IFN) [40]. The regulation of cDAMPs and iDAMPs, predominantly at posttranslational level, has been well described. However, while it is well known that miRNAs are important for the regulation of cancer immunoediting, little is known about the potential modulation of miRNAs by ICD on cancer cells and their eventual impact in the induction of antitumor-specific immune responses [4,5,6].

In this study, we explored the participation of miRNAs in ICD using two well-established models of ICD, in particular, Mino cells treated with the RA/IFNα combination [20,31] and MDA-MB-231 cells exposed to DXR [41]. 

miRNA are short, noncoding RNAs whose dysregulation has been involved in tumors, hence the identification of factors controlling miRNA expression for the treatment of cancer is an area of growing interest. miRNAs can act as oncomiRNAs or tumor suppressor miRNAs by directly targeting tumor suppressive mRNAs and oncogenic mRNAs, respectively [42]. We have recently recapitulated current knowledge of cancer-associated miRNAs targeting DAMPs transcripts, through which new molecular mediators that account for ICD immunogenicity were identified [17]. Surprisingly, we found that there is little information regarding miRNAs modulation during ICD, even though different miRNAs were shown to contribute to the therapeutic efficacy of ICD inducers [43,44]. 

Herein, analysis of miRNA expression profiles identified 16 miRNAs differentially expressed in cells subjected to ICD relative to viable and gamma-irradiated/necrotic cells. Intriguingly, the target genes of these differentially expressed miRNAs were enriched in important biological pathways associated with Immune System Process (GO Term) indicating the involvement of these specific miRNAs in immunomodulatory features of ICD. 

Highly and differentially expressed mRNAs identified in this study suggested a new immunomodulatory activity of ICD inducers resulting in the regulation of antigen presentation, a crucial step for proper activation of the immune system antitumor response. Indeed, enrichment analysis demonstrated that the MHC-I pathway was upregulated, as previously reported [45,46], whereas MHC-II was downregulated. An in silico analysis of publicly available database [16] showed high concordance with our results, generalizing our findings, especially those regarding the downregulation of class II MHC molecules, an event associated to ICD induction altogether rather than to the specific ICD-inducing anticancer treatment.

MHC-II is a heterodimer consisting of an alpha and a beta chain. In humans, there are multiple MHC-II isotypes: HLA-DR, HLA-DP, and HLA-DQ [47]. Multiple transcriptional and post-transcriptional mechanisms are implicated in the regulation of the expression of MHC-II molecules. Generally, genes participating in MHC-II-dependent antigen presentation are co-regulated by the Class II transactivator (CIITA). CIITA binds to transcriptional complexes involved in constitutive and IFN-γ-induced Class II MHC expression. Once expressed, MHC-II alpha and beta chains are assembled in a complex with MHC-II-associated invariant chain (Ii, CD74) in the endoplasmic reticulum (ER) to avoid irregular peptide loading within the ER. Then, MHC-II is transported via vesicles that sprout from the ER and Ii is responsible for directing MHC-II-containing vesicles to acidic endosomes, where MHC-II is loaded with antigen peptides. Binding of peptide antigen is catalyzed by co-chaperones, such as HLA-DO or HLA-DM. MHC-II complexes are stable only when antigen-loaded, and importantly, only stable MHC-II complexes persist at the cell surface. Once on the cell surface, antigenic fragments presented by MHC-II complexes can be recognized primarily by CD4+ T cells [48]. 

Upregulation of MHC-II expression by cancer cells has been reported in different human tumors including many solid tumors where the tissue of origin does not normally express MHC-II molecules: classic Hodgkin lymphoma, colorectal cancer, prostate cancer, breast cancer, ovarian cancer, glioma, melanoma, and non-small cell lung cancer [48,49,50,51,52,53,54]. MHC-II expression by cancer cells has an important role in antitumor immunity. Accordingly, independent studies showed an association between tumor-expressed MHC-II, enhanced antitumor immunity, and more favorable prognosis [50,51,52,55]. Tumor-expressed MHC-II also correlated with response to anti-PD-1/anti-PD-L1 [49,56]. However, contrasting reports indicated that MHC-II expression in tumors correlated with poor survival [57,58]. Along this line, the MHC-II is also a known ligand of the inhibitory co-receptor LAG3, and their interaction provides immunosuppressive signals [59]; its upregulation in tumor-infiltrating lymphocytes involved LAG3 as a key immune checkpoint in the tumor microenvironment [60]. LAG3 is mostly found on conventional CD4+ and CD8+ T cells, T regulatory (Treg) cells, NK cells, and plasmacytoid dendritic cells, displaying significant immune regulatory properties dependent on interaction with diverse ligands, including but not limited to MHC-II [61]. Nevertheless, by interacting with MHC-II, LAG3 on Treg cells negatively regulates CD4+ T cells proliferation and inhibits cytokine responses [59] along with suppressing DC expansion and maturation [62]. LAG3 on CD8+ T cells can inhibit their activation in a manner dependent on the density of MHC-II on APCs [59]. Interestingly, this interaction also affected antitumor immunity in MHC-II-expressing melanoma cells, whereby these were protected from apoptosis promoted by LAG3-overexpressing T cells [63]. Moreover, LAG3 is not the only inhibitory checkpoint involved in the pro-tumoral function of tumor-expressed MHC-II. Fc receptor-like 6 (FCRL6), another MHC-II ligand involved in adaptive resistance to immunotherapy, delivers an immunosuppressive signal that directly represses NK cell cytotoxic activity and effector T cell cytokine secretion [64].

In this study, supporting evidence was derived from analysis of MHC-II-associated transcripts and HLA-DR surface expression on lymphoma cells subjected to ICD, which validated the downregulating effect of ICD on the MCH-II antigen presentation pathway. Considering the involvement of MHC-II in mediating anti-tumor CD4+ T cells responses, its downregulation by ICD could play an undesirable role in the modulation of antitumor immune responses [48]. On the other hand, reduced engagement of the inhibitory co-receptors LAG3, a novel immunotherapeutic target in cancer and particularly B lymphomas, or FCRL6, due to decreased expression of its MHC-II ligand, could contribute to explaining the enhancement of anticancer immune responses by ICD.

To the best of our knowledge, this is the first study that investigates miRNA expression profiles to identify the miRNA signature associated with ICD-promoted events. A recent report demonstrated that specific miRNA clustering patterns would likely result from specific cell stimuli, e.g., antitumor treatments, which could be employed in each instance to modulate specific responses for therapeutic purposes [65]. In this report, we have identified modules of co-expressed miRNAs regulated by ICD which would allow repression of immune-associated signaling pathways. miR-4284 and miR-212-3p were the miRNAs most strongly upregulated by ICD. Based on their seed sequence, those miRNAs were the main candidate that might regulate key genes involved in antigen presentation pathways. Several reports have documented the aberrant expression of miR-212-3p [65,66,67,68] and miR-4284 [69] in various tumors in comparison with paired normal tissues. These studies postulated that detection of these miRNAs may have diagnostic or prognostic value in several neoplastic diseases. In support of this notion, their expression has been linked to several hallmarks of cancer development and progression: tumorigenicity, proliferation, migration, invasion, angiogenesis, and therapy-promoted cell death [70,71,72,73,74,75,76,77,78,79,80]. In contrast, no information is available regarding their involvement in ICD. Here, we demonstrate that miR-212-3p contributes to downregulated surface expression of HLA-DR, thus affecting immune system regulation. Our data are in accordance with previous evidence showing that miR-212-3p inhibited MHC II expression in pancreatic cancer cells [81], which transferred this blockade to iDC via exosomes [82]. 

When studying miRNAs, it is important to consider the relative promiscuity of their targets: a given miRNA may have thousands of targets with significant differences in function and, conversely, a single gene could be targeted by different miRNAs [35]. In this context, it seems logical to propose that miR-4284 and miR-212-3p could be acting in a synergistic way, thus preventing compensatory regulatory mechanisms that could counteract miRNA-mediated gene silencing. In line with this, clinical trial data demonstrated that employing single miRNA mimics may have limited therapeutic impact, suggesting that multiple miRNAs could coordinate in the regulation of complex cellular pathways, thus exerting more extensive biological effects [83,84]. Further functional studies are required to clarify the combined effects of multiple miRNAs in the regulation of the MHC-II pathway. 

## 5. Conclusions

Overall, through comprehensive analysis and validation, this study uncovered that the miRNA profile was significantly altered during ICD. Several miRNAs, in particular miR-212-3p, are predicted to be involved in the regulation of the antigen presentation pathway, whose implication in ICD is demonstrated herein for the first time. Our findings provide useful insights to improve the design of combination immunotherapies exploiting the immune-related therapeutic properties of ICD inducers.

## Figures and Tables

**Figure 1 biomedicines-10-01896-f001:**
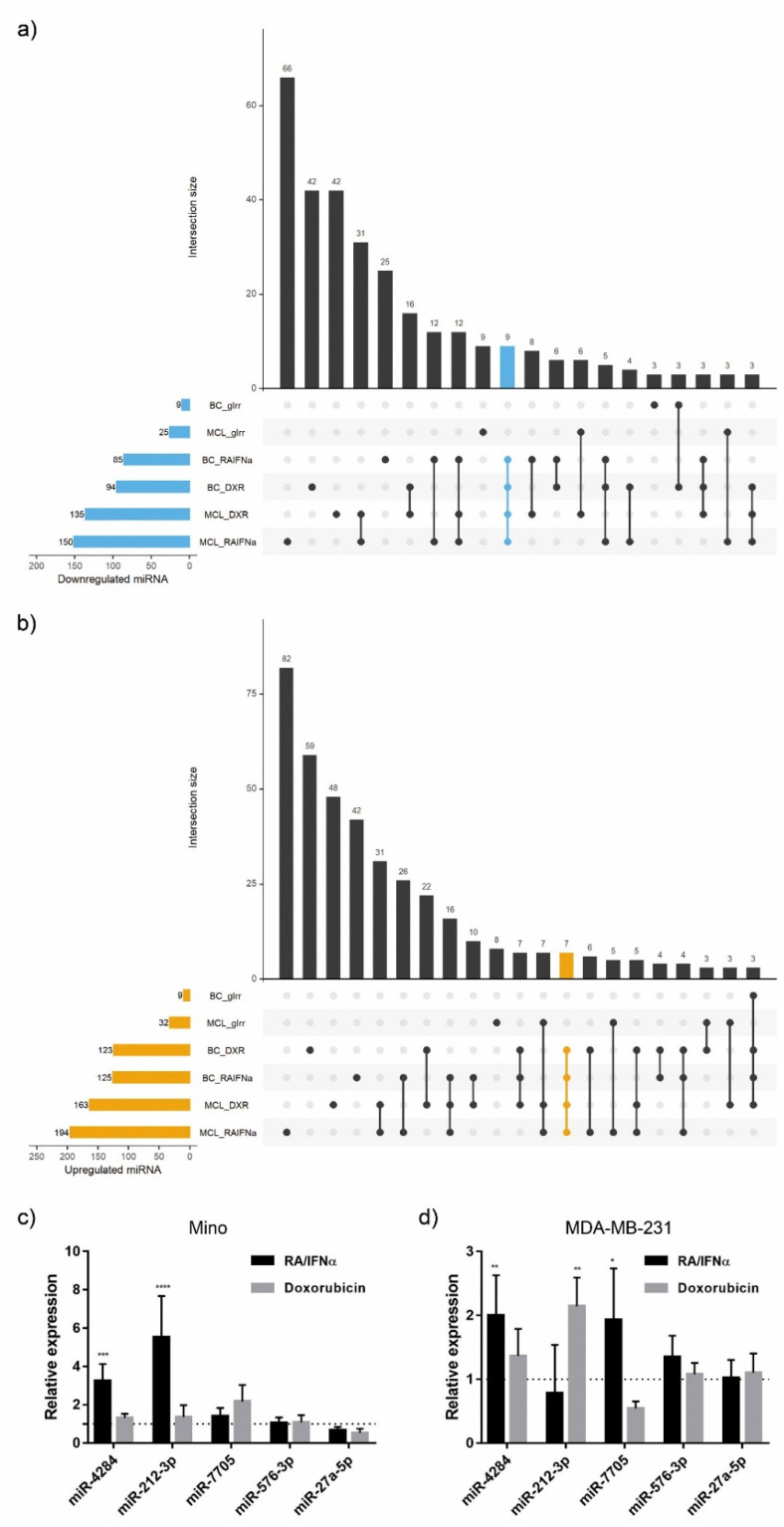
**miRNA modulation during immunogenic cell death.** Mantle cell lymphoma (MCL: Mino) and breast cancer (BC: MDA-MB-231) cell lines were treated with two ICD-inducers: 9-cis Retinoic Acid and Interferon-alpha (RA/IFNα) and doxorubicin (DXR), or exposed to not-ICD inducing treatment with gamma irradiation (gIrr). RNA-seq was performed through NGS technology. (**a**,**b**) UpSet plots of miRNAs differentially downregulated (**a**) and upregulated (**b**) between control and treatment. The significance of miRNA expression changes was inferred based on an adjusted *p*-value < 0.05 and FC ≤ −1.3 and FC ≥ 1.3, respectively. The blue (**a**) and orange (**b**) horizontal bars on the left show the number of differentially expressed miRNAs identified by each treatment vs. untreated cells, ordered by set size. The vertical bars display the size of sets of miRNAs shared between comparisons and the intersection sets. The interaction between ICD inducers excluding not-ICD treatment is highlighted in blue (**a**) or orange (**b**). (**c**,**d**) RT-qPCR based quantification of miRNAs expression in MCL and BC cell lines treated with RA/IFNα and doxorubicin. Expression values were normalized to U6 snRNA level and referred to control untreated samples (dotted line = 1). Statistical analysis (treated vs. untreated cells): Two-way ANOVA, Dunnett’s multiple comparisons test, * *p* < 0.05; ** *p* < 0.01; *** *p* < 0.001, **** *p* < 0.0001.

**Figure 2 biomedicines-10-01896-f002:**
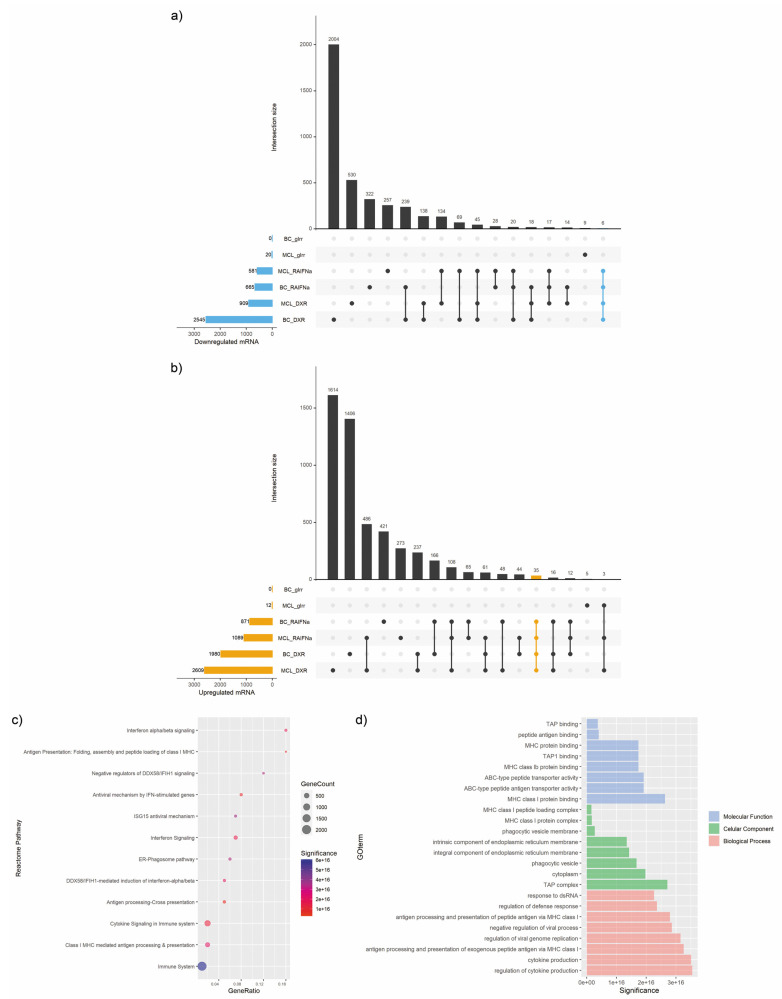
**mRNA modulation during immunogenic cell death.** Mantle cell lymphoma (MCL: Mino) and breast cancer (BC: MDA-MB-231) cell lines were treated with two ICD-inducers: 9-cis Retinoic Acid and Interferon-alpha (RA/IFNα) and doxorubicin (DXR) or exposed to not-ICD inducing treatment with gamma irradiation (gIrr). RNA-seq was performed through NGS technology. (**a**,**b**) UpSet plots of mRNA differentially downregulated (**a**) and upregulated (**b**) between control and treatments. The significance of gene expression changes was inferred based on an adjusted *p*-value < 0.5 and FC ≤ −1.3 (**a**) or FC ≥ 1.3 (**b**). The blue (**a**) and orange (**b**) horizontal bars on the left show the number of differentially expressed genes identified by each treatment vs. untreated cells, ordered by set size. The vertical bars display the size of sets of genes shared between comparisons and the intersection sets. The interaction between ICD inducers excluding not-ICD inducing treatment is highlighted in blue (**a**) or in orange (**b**). (**c**) Bubble plots showing overrepresented Reactome pathways in upregulated genes by ICD inducers. The *y*-axis represents the significantly enriched pathways, the *x*-axis represents the gene ratio (intersection size/term size), the size of the bubbles represents the number of assigned genes, and the color of bubbles represents the significance (negative log10 *p*-value). The larger number of genes classified into the pathway, the larger the node size is. (**d**) Bar plot of enriched gene ontology (GO) terms of upregulated genes by ICD. *Y*-axis indicates the GO term, *x*-axis indicates the significance (negative log10 *p*-value).

**Figure 3 biomedicines-10-01896-f003:**
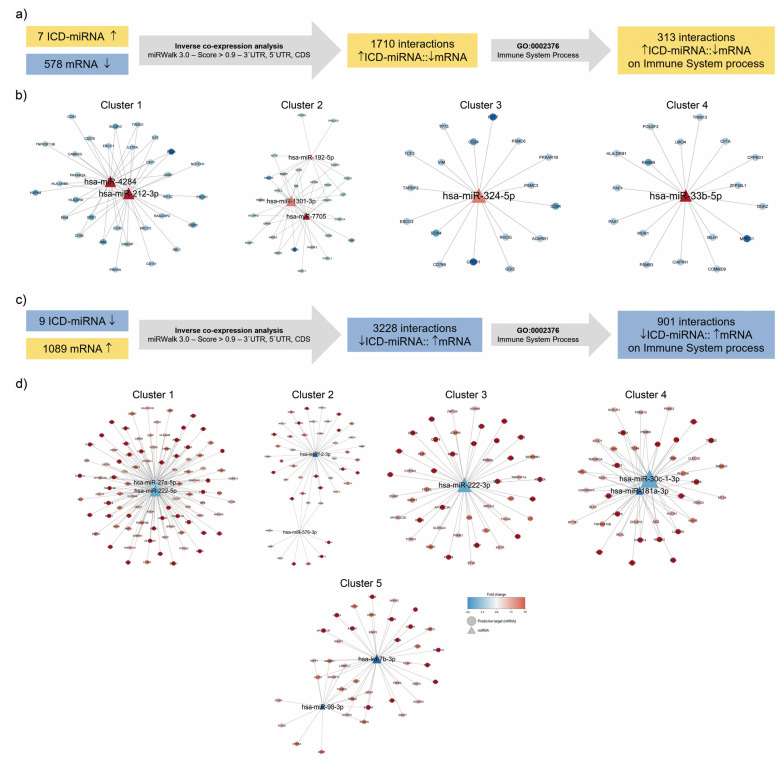
**Clustering of immune miRNA: mRNA pairs.** (**a**) Identification of inverse co-expression relationship between miRNAs whose expression level was upregulated by ICD inducers, and predictive immune system process-associated targets downregulated by RA/IFNα. (**b**) Clustering analysis was conducted on upregulated ICD-miRNA::mRNA pair. (**c**) Identification of inverse co-expression relationship between miRNAs whose expression level was downregulated by ICD, and predictive immune system process-associated targets upregulated by RA/IFNα. (**d**) Clustering analysis was conducted on downregulated ICD-miRNA::mRNA pair.

**Figure 4 biomedicines-10-01896-f004:**
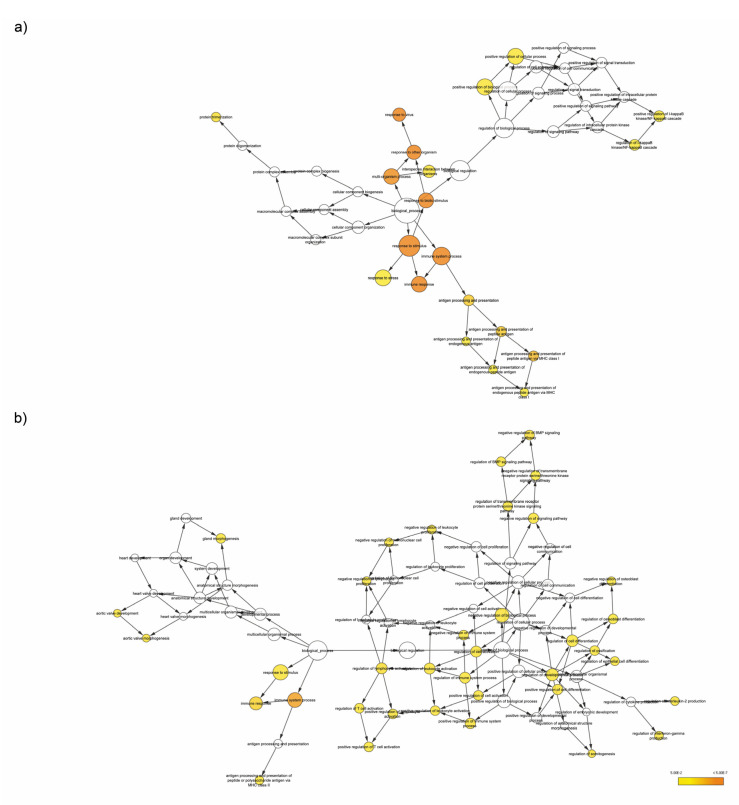
**Pathway enrichment analysis.** (**a**) Enrichment analysis of genes upregulated by RA/IFNα and predicted targets of miR-27-5a-p and miR-222-5p (Cluster 1) was conducted using the BiNGO app in Cytoscape. (**b**) Enrichment analysis of genes downregulated by RA/IFNα and predicted targets of miR-4284 and miR-212-3p (Cluster 1) was conducted using the BiNGO app in Cytoscape.

**Figure 5 biomedicines-10-01896-f005:**
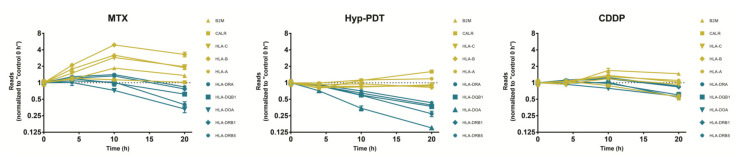
**Expression profile of Class I- and Class II-associated transcripts.** Results of meta-analyses of data published in the GEO database (GEO ID: GSE163377), regarding Class I (B2M, CALR, HLA-C, HLA-B, HLA-A) or Class II (HLA-DRA, HLA-DQB1, HLA-DOA, HLA-DRB1, HLA-DRB5) MHC-associated transcripts. These gene sets include the RNA-seq data of A375 melanoma cells responding to mitoxantrone (MTX) or hypericin-based photodynamic therapy (Hyp-PDT), as immunogenic treatments, and cisplatin (CDDP) as an experimental model of non-ICD inducer, after 0, 4, 10, and 20 h. Reads were normalized to values corresponding to control at 0 h.

**Figure 6 biomedicines-10-01896-f006:**
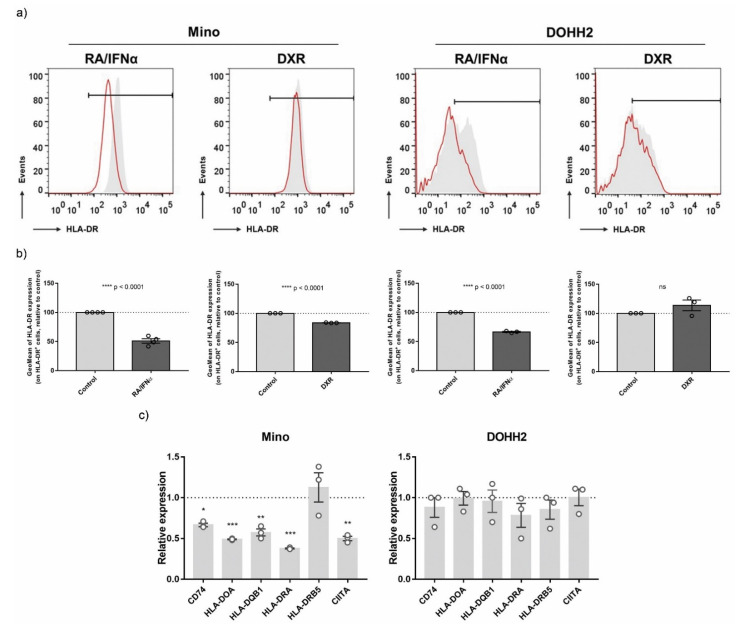
**Class II MHC antigen presentation pathway****modulation by ICD.** (**a**) HLA-DR (MHC-II) surface expression was validated by flow cytometry in control (grey untreated) and RA/IFNα- (left) and DXR-treated (right) (red line ICD inducers) on B-cell lymphoma cells (Mino and DOHH2). Representative histograms are shown. (**b**) HLA-DR surface expression values (GeoMean) were normalized to control untreated samples (dotted line = 1). Statistical analysis (treated vs. untreated cells): *t*-test, **** *p* < 0.0001, ns = non-significant. (**c**) Class II MHC-associated transcripts were evaluated by RT-qPCR in control (untreated) and RA/IFNα (immunogenic cell death inducer)-treated B-cell lymphoma cell lines: Mino (left) and DOHH2 (right). Expression values were normalized to GAPDH level and referred to control untreated samples (dotted line = 1). Statistical analysis (treated vs. untreated cells): one-way ANOVA, Dunnett’s multiple comparisons test, * *p* < 0.05; ** *p* < 0.01; *** *p* < 0.001.

**Figure 7 biomedicines-10-01896-f007:**
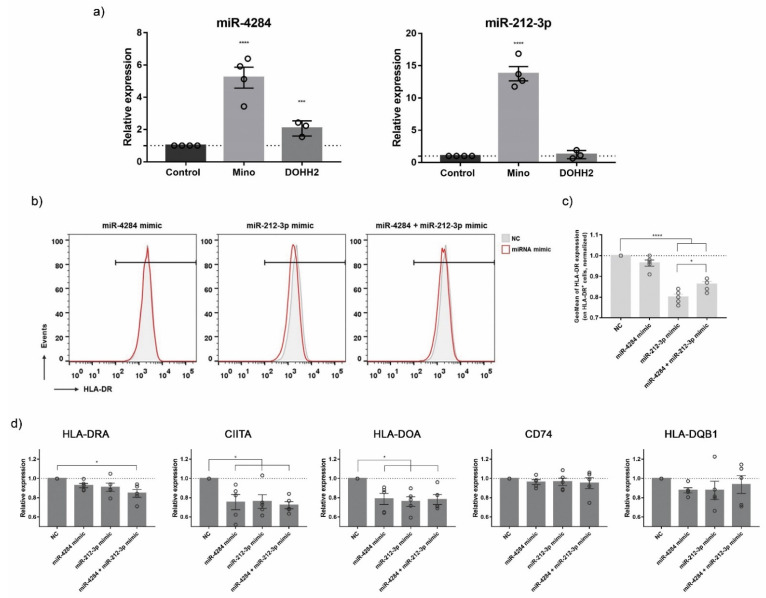
**Class II MHC antigen presentation pathway modulation by miRNAs upregulated by ICD.** (**a**) miR-4284 (left) and miR-212-3p (right) expression was evaluated by RT-qPCR in control (untreated) and RA/IFNα (ICD inducer)-treated cell lines: Mino and DOHH2. Expression values were normalized to U6 snRNA levels and referred to control untreated samples (dotted line = 1). Statistical analysis (treated vs. untreated cells): one-way ANOVA, Dunnett’s multiple comparisons test, *** *p* < 0.001, **** *p* < 0.0001. (**b**) HLA-DR (MHC-II) surface expression was validated by flow cytometry in negative control-transfected cells (grey NC) and miR-4284, miR-212-3p or both miRNAs mimic-transfected cells (red line miRNA mimic) on Meljuso cells. Representative histograms are shown. (**c**) HLA-DR surface expression values (GeoMean) were normalized to negative control-transfected samples (dotted line = 1). Statistical analysis: one-way ANOVA, Dunnett’s multiple comparisons test, * *p* < 0.05; **** *p* < 0.0001. (**d**) Class II MHC -associated transcripts were evaluated by RT-qPCR in negative control-transfected cells and miR-4284, miR-212-3p or both miRNAs mimic-transfected cells on Meljuso cells. Expression values were normalized to GAPDH levels and referred to control untreated samples (dotted line = 1). Statistical analysis: one-way ANOVA, Dunnett’s multiple comparisons test, * *p* < 0.05.

**Table 1 biomedicines-10-01896-t001:** **Differentially expressed miRNAs in immunogenic cell death.** miRNAs modulated by ICD inducers were identified through the selection of miRNAs whose expression was significantly different in DXR- or RA/IFNα-treated samples versus untreated ones (*p* ≤ 0.05, FC ≤ −1.3 for downregulated, FC ≥ 1.3 for upregulated), but not significantly different (NS) in gamma irradiation-treated samples. Downregulated miRNAs are highlighted in blue, upregulated miRNAs are highlighted in orange.

ID	miRBase Accesion	Fold Change						*p*-Value					
		Mino			MDA-MB-231			Mino			MDA-MB-231		
		gIrr vs. Untreated	DXR vs. Untreated	IFNα/RA vs. Untreated	gIrr vs. Untreated	DXR vs. Untreated	IFNα/RA vs. Untreated	gIrr vs. Untreated	DXR vs. Untreated	IFNα/RA vs. Untreated	gIrr vs. Untreated	DXR vs. Untreated	IFNα/RA vs. Untreated
hsa-let-7b-3p	MIMAT0004482	NS	−2.37	−3.26	NS	−1.97	−1.68	NS	1.20 × 10^−70^	1.07 × 10^−137^	NS	7.71 × 10^−58^	1.14 × 10^−34^
hsa-let-7f-2-3p	MIMAT0004487	NS	−1.66	−3.71	NS	−1.42	−1.45	NS	3.93 × 10^−43^	1.10 × 10^−233^	NS	1.56 × 10^−24^	5.83 × 10^−26^
hsa-miR-576-3p	MIMAT0004796	NS	−1.44	−1.89	NS	−1.38	−1.38	NS	4.30 × 10^−2^	4.28 × 10^−5^	NS	2.41 × 10^−2^	2.77 × 10^−2^
hsa-miR-98-3p	MIMAT0022842	−1.25	−1.47	−22.34	NS	−1.34	−2.29	1.08 × 10^−4^	1.67 × 10^−14^	1.00 × 10^−169^	NS	6.61 × 10^−17^	1.04 × 10^−102^
hsa-miR-30c-1-3p	MIMAT0004674	−1.15	−2.20	−2.40	NS	−1.60	−1.79	1.36 × 10^−2^	5.77 × 10^−60^	8.42 × 10^−85^	NS	4.11 × 10^−23^	5.49 × 10^−32^
hsa-miR-181a-3p	MIMAT0000270	−1.14	−2.46	−3.71	NS	−1.78	−1.52	6.43 × 10^−18^	0.00	0.00	NS	0.00	1.01 × 10^−165^
hsa-miR-27a-5p	MIMAT0004501	−1.12	−1.88	−1.89	NS	−1.55	−1.43	4.81 × 10^−2^	3.57 × 10^−46^	2.91 × 10^−56^	NS	6.96 × 10^−46^	1.75 × 10^−30^
hsa-miR-222-3p	MIMAT0000279	1.09	−1.32	−2.14	−1.06	−1.54	−2.26	1.37 × 10^−3^	2.89 × 10^−35^	7.25 × 10^−256^	4.89 × 10^−2^	8.52 × 10^−96^	0.00
hsa-miR-222-5p	MIMAT0004569	2.03	−4.15	−2.94	1.12	−2.08	−2.34	2.11 × 10^−65^	8.41 × 10^−84^	5.46 × 10^−84^	1.39 × 10^−8^	0.00	0.00
hsa-miR-212-3p	MIMAT0000269	NS	3.59	48.61	NS	1.70	1.56	NS	4.89 × 10^−3^	3.21 × 10^−28^	NS	1.28 × 10^−6^	1.02 × 10^−4^
hsa-miR-4284	MIMAT0016915	NS	6.08	11.55	NS	1.33	1.70	NS	8.03 × 10^−15^	2.08 × 10^−29^	NS	7.29 × 10^−3^	2.05 × 10^−8^
hsa-miR-33b-5p	MIMAT0003301	NS	1.71	7.66	NS	1.34	7.36	NS	1.90 × 10^−36^	0.00	NS	6.39 × 10^−13^	0.00
hsa-miR-7705	MIMAT0030020	NS	2.44	6.48	NS	1.70	3.30	NS	1.57 × 10^−7^	1.12 × 10^−39^	NS	1.98 × 10^−38^	1.70 × 10^−203^
hsa-miR-1301-3p	MIMAT0005797	NS	1.59	2.78	NS	1.85	1.64	NS	9.14 × 10^−21^	2.66 × 10^−121^	NS	5.24 × 10^−13^	3.32 × 10^−8^
hsa-miR-324-5p	MIMAT0000761	NS	1.50	2.40	NS	1.30	1.39	NS	7.82 × 10^−11^	4.45 × 10^−57^	NS	3.33 × 10^−7^	2.21 × 10^−10^
hsa-miR-192-5p	MIMAT0000222	NS	1.67	1.51	NS	2.44	1.43	NS	9.77 × 10^−229^	2.97 × 10^−149^	NS	0.00	7.08 × 10^−120^

**Table 2 biomedicines-10-01896-t002:** **Differentially expressed mRNAs during immunogenic cell death.** mRNAs modulated by ICD inducers were identified selecting those whose expression was significantly different in DXR- or RA/IFNα-treated samples versus untreated ones (adjusted *p*-value ≤ 0.05, FC ≤ −1.3 for downregulated, FC ≥ 1.3 for upregulated), but not significantly different in gamma irradiation-treated samples. Downregulated mRNAs are highlighted in blue, upregulated mRNAs are highlighted in orange.

ID	Fold Change						*p*-Value					
	Mino			MDA-MB-231			Mino			MDA-MB-231		
	gIrr vs. Untreated	DXR vs. Untreated	IFNα/RA vs. Untreated	gIrr vs. Untreated	DXR vs. Untreated	IFNα/RA vs. Untreated	gIrr vs. Untreated	DXR vs. Untreated	IFNα/RA vs. Untreated	gIrr vs. Untreated	DXR vs. Untreated	IFNα/RA vs. Untreated
DPYSL2	1.10	−1.69	−1.89	−1.01	−3.84	−2.84	1.00	1.81 × 10^−2^	3.63 × 10^−3^	1.00	2.91 × 10^−^^14^	2.00 × 10^−^^8^
ENC1	1.14	−1.52	−2.35	−1.04	−1.46	−1.53	1.00	1.61 × 10^−^^2^	7.31 × 10^−^^8^	1.00	4.42 × 10^−^^3^	2.09 × 10^−^^3^
MEGF9	−1.36	−1.36	−1.33	−1.04	−1.80	−1.50	1.78 × 10^−^^1^	1.74 × 10^−^^2^	2.70 × 10^−^^2^	1.00	1.48 × 10^−^^10^	7.77 × 10^−^^5^
NIN	−1.12	−1.42	−1.38	1.05	−1.55	−1.74	1.00	4.30 × 10^−^^3^	8.08 × 10^−^^3^	1.00	1.66 × 10^−^^5^	7.53 × 10^−^^8^
PPM1F	1.15	−1.44	−1.42	−1.02	−1.86	−1.48	1.00	1.56 × 10^−^^2^	2.28 × 10^−^^2^	1.00	7.03 × 10^−^^8^	3.48 × 10^−^^3^
TTC3	−1.05	−1.49	−1.39	1.02	−1.50	−1.65	1.00	1.22 × 10^−^^2^	4.92 × 10^−^^2^	1.00	4.65 × 10^−^^3^	5.93 × 10^−^^4^
ADM	−1.05	1.35	1.40	−1.02	2.68	2.02	1.00	2.70 × 10^−^^2^	1.47 × 10^−^^2^	1.00	4.98 × 10^−^^29^	9.12 × 10^−^^14^
ANKRD33B	−1.02	1.34	1.49	1.15	2.45	1.67	1.00	2.54 × 10^−^^2^	1.29 × 10^−^^3^	1.00	3.86 × 10^−^^16^	4.89 × 10^−^^5^
CEACAM1	−2.06	4.19	6.52	2.92	9.15	26.13	NS	4.04 × 10^−^^3^	2.62 × 10^−^^5^	1.00	6.84 × 10^−^^3^	3.32 × 10^−^^5^
DDX58	1.02	9.96	10.85	1.05	2.66	6.46	1.00	2.96 × 10^−^^6^	1.49 × 10^−^^7^	1.00	3.94 × 10^−^^2^	2.13 × 10^−^^5^
HERC5	−1.45	149.92	213.69	1.06	3.60	5.23	1.00	1.61 × 10^−^^5^	3.92 × 10^−^^7^	1.00	8.48 × 10^−^^3^	7.83 × 10^−^^4^
HLA-B	1.03	2.12	2.47	1.04	1.60	2.95	1.00	2.98 × 10^−^^4^	9.62 × 10^−^^7^	1.00	1.50 × 10^−^^2^	5.89 × 10^−^^10^
HLA-E	1.04	2.61	2.52	1.04	2.32	2.21	1.00	2.62 × 10^−^^3^	2.66 × 10^−^^3^	1.00	2.84 × 10^−^^3^	1.10 × 10^−^^2^
IFI27L1	1.11	1.42	1.58	−1.07	1.42	1.40	1.00	3.11 × 10^−^^2^	2.98 × 10^−^^3^	1.00	5.86 × 10^−^^3^	2.19 × 10^−^^2^
IFI44	−1.61	39.21	3133.84	1.09	5.48	18.69	1.00	7.36 × 10^−^^3^	6.20 × 10^−^^13^	1.00	2.05 × 10^−^^2^	2.54 × 10^−^^5^
IFI6	1.25	66.39	135.73	1.12	7.39	112.41	1.00	2.86 × 10^−^^9^	2.19 × 10^−^^14^	1.00	1.47 × 10^−^^3^	2.61 × 10^−^^16^
IFIH1	−1.11	6.94	11.16	1.06	3.06	15.55	1.00	2.04 × 10^−^^5^	2.76 × 10^−^^9^	1.00	9.59 × 10^−^^3^	1.31 × 10^−^^12^
IFIT1	−1.60	374.39	2665.15	1.03	5.84	27.03	1.00	2.96 × 10^−^^6^	2.60 × 10^−^^12^	1.00	1.74 × 10^−^^2^	1.90 × 10^−^^6^
IFIT2	1.07	99.54	140.41	1.02	7.65	5.51	1.00	6.41 × 10^−^^7^	4.33 × 10^−^^9^	1.00	3.48 × 10^−^^3^	3.48 × 10^−^^2^
IFIT3	−1.02	21.47	81.05	−1.04	4.85	10.30	1.00	2.45 × 10^−^^7^	8.54 × 10^−^^17^	1.00	2.30 × 10^−^^3^	4.08 × 10^−^^6^
ISG15	1.15	11.46	29.74	1.09	2.62	12.29	1.00	2.53 × 10^−^^7^	5.05 × 10^−^^16^	1.00	4.01 × 10^−^^2^	1.02 × 10^−^^9^
ISG20	−1.24	7.94	14.92	1.36	6.74	16.63	1.00	3.36 × 10^−^^5^	9.48 × 10^−^^10^	1.00	6.96 × 10^−^^5^	1.79 × 10^−^^9^
MDK	1.02	5.67	5.04	1.03	2.83	2.92	1.00	1.03 × 10^−^^3^	1.33 × 10^−^^3^	1.00	2.44 × 10^−^^2^	3.82 × 10^−^^2^
MX2	1.00	3.66	6.58	−1.27	2.30	40.51	1.00	1.99 × 10^−^^5^	1.26 × 10^−^^12^	1.00	7.06 × 10^−^^3^	1.79 × 10^−^^45^
OASL	−1.50	426.45	710.61	−1.04	11.42	10.78	1.00	9.62 × 10^−^^7^	5.52 × 10^−^^9^	1.00	1.86 × 10^−^^4^	6.20 × 10^−^^4^
PELI1	1.04	1.38	1.43	−1.05	1.63	1.47	1.00	1.23 × 10^−^^2^	4.10 × 10^−^^3^	1.00	4.44 × 10^−^^4^	2.15 × 10^−^^2^
PIM2	−1.05	1.34	1.56	−1.01	1.67	2.39	1.00	2.63 × 10^−^^2^	2.73 × 10^−^^4^	1.00	5.06 × 10^−^^5^	6.56 × 10^−^^13^
PLEKHA4	−1.34	14.67	51.65	−1.23	3.87	3.33	NS	2.57 × 10^−^^2^	1.42 × 10^−^^4^	1.00	1.86 × 10^−^^10^	9.16 × 10^−^^8^
PNPLA2	−1.10	1.55	1.62	−1.01	1.56	1.49	1.00	1.73 × 10^−^^2^	4.45 × 10^−^^3^	1.00	1.09 × 10^−^^10^	9.63 × 10^−^^8^
PSMB9	1.07	1.91	3.05	1.10	1.55	2.36	1.00	1.39 × 10^−^^3^	2.48 × 10^−^^11^	1.00	1.12 × 10^−^^2^	6.13 × 10^−^^8^
RNF114	1.18	1.53	1.31	1.01	1.45	1.49	1.00	5.61 × 10^−^^4^	3.88 × 10^−^^2^	1.00	2.04 × 10^−^^4^	1.56 × 10^−^^4^
RSAD2	−3.26	41.51	214.37	−2.47	18.77	491.58	NaN	5.07 × 10^−^^3^	4.02 × 10^−^^6^	1.00	1.84 × 10^−2^	2.40 × 10^−^^8^
RTP4	−1.23	4.07	4.89	−1.12	8.17	43.22	NaN	4.34 × 10^−^^3^	5.59 × 10^−^^4^	1.00	1.66 × 10^−^^3^	1.02 × 10^−^^9^
SAMD9	−1.14	8.24	6.89	−1.09	3.14	6.83	1.00	3.60 × 10^−^^5^	6.22 × 10^−^^5^	1.00	2.01 × 10^−^^2^	4.13 × 10^−^^5^
SCAMP1-AS1	1.03	2.26	2.30	1.01	2.04	3.42	NS	2.12 × 10^−^^2^	2.40 × 10^−^^2^	1.00	4.89 × 10^−^^2^	2.82 × 10^−^^4^
SRGN	−1.08	1.77	1.71	−1.03	1.54	1.66	1.00	6.71 × 10^−^^3^	1.07 × 10^−^^2^	1.00	2.87 × 10^−^^2^	1.60 × 10^−^^2^
ST3GAL1	−1.08	1.40	1.51	1.07	1.40	1.53	1.00	4.71 × 10^−^^3^	1.67 × 10^−^^4^	1.00	8.86 × 10^−^^4^	4.33 × 10^−^^5^
TAP1	1.04	3.09	4.10	1.05	2.12	3.53	1.00	2.05 × 10^−^^5^	2.42 × 10^−^^9^	1.00	1.99 × 10^−^^3^	5.53 × 10^−^^8^
TAP2	−1.24	1.34	2.30	−1.01	1.93	1.46	9.03 × 10^−^^1^	2.55 × 10^−^^2^	3.75 × 10^−^^15^	1.00	3.22 × 10^−^^14^	1.60 × 10^−^^4^
ZNF600	1.07	1.84	1.55	−1.00	1.76	1.56	1.00	4.32 × 10^−^^3^	5.28 × 10^−^^2^	1.00	1.92 × 10^−^^3^	4.58 × 10^−^^2^
ZNFX1	−1.04	2.05	2.84	−1.01	1.56	2.91	1.00	2.34 × 10^−^^3^	3.73 × 10^−^^7^	1.00	4.48 × 10^−^^2^	4.86 × 10^−^^8^

**Table 3 biomedicines-10-01896-t003:** **Cluster generation of upregulated ICD-miRNA: mRNA pairs.** Nodes: Vertices of the network (here, miRNAs and genes). Edges: Connections between the nodes interactions within the network (here, miRNA:mRNA interaction). Modularity: Comparison between the observed and expected number of edges between nodes in the same cluster.

Cluster	miRNAs	Nodes	Edges	Modularity
1	hsa-miR-4284 hsa-miR-212-3p	33	53	0.482
2	hsa-miR-192-5p hsa-miR-1301-3p hsa-miR-7705	29	23	0.53
3	hsa-miR-324-5p	19	18	0.277
4	hsa-miR-33b-5p	18	17	0.246

**Table 4 biomedicines-10-01896-t004:** **Cluster generation of downregulated ICD-miRNA: mRNA pair.** Nodes: Vertices of the network (here, miRNAs and genes). Edges: Connections between the nodes interactions within the network (here, miRNA:mRNA interaction). Modularity: Comparison between the observed and expected number of edges between nodes in the same cluster.

Cluster	miRNAs	Nodes	Edges	Modularity
1	hsa-miR-27a-5p hsa-miR-222-5p	93	149	0.466
2	hsa-miR-576-3p hsa-let-7f-2-3p	48	50	0.314
3	hsa-miR-222-3p	46	45	0.250
4	hsa-miR-30c-1-3p hsa-miR-181a-3p	43	71	0.283
5	hsa-miR-98-3p hsa-let-7b-3p	42	51	0.319

## Data Availability

Raw RNA sequencing data and smallRNA-Seq data are deposited in the EBI ArrayExpress database (http://www.ebi.ac.uk/arrayexpress) with accession numbers E-MTAB-11799 and E-MTAB-11803, respectively (accessed on 7 April 2022).

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
