# Peer review of "Integration of miRNA:mRNA Co-Expression Revealed Crucial Mechanisms Modulated in Immunogenic Cancer Cell Death"

_biomedicines, 2022, doi:10.3390/biomedicines10081896_

Round 1

Reviewer 1 Report

Lamberti et al. have studied the changes induced during Immunogenic Cell Death (ICD) in miRNA and mRNA profiles by next generation sequencing. Their results show that 16 miRNAs were differentially modulated (9 downregulated and 7 upregulated) in lymphoma and breast cancer cells treated with the combination of retinoic acid (RA) and interferon-alpha or doxorubicin. Upregulation of miR-4284 and miR-212-3p was confirmed by by RT-qPCR, although the effect was not observed in some of the treatments. RNA-seq analyses revealed that 6 genes were downregulated by ICD inducers  and 35 genes were upregulated. Among the genes regulated by ICD inducers, the authors found a possible connection between miR-4284 and miR-212-3p upregulation and MCH-II downregulation. Moreover, the authors propose that downregulation of MCH-II  could be a feature of ICD. Finally, expression of MHC-II after RA/IFN-alpha has been determined by flow cytometry. A modest reduction in the MFI suggest downregulation of MHC-II.

In my opinion, these are the main issues that should be addressed before it can be accepted for publication:

1) Apparently the doses of ICD-inducers were the same for the lymphoma and breast cancer cell lines. In a previous study of the same laboratory the authors reported that RA/IFN-alpha induces ICD in lymphoma cell lines, but there are not available data about sensitivity the immunogeneicity of this combination in MDA-MB-231 cells. Experiments demonstrating that RA/IFN-alpha induces ICD in MDA-MB-231 should be included.

2) Gamma-irradiation (gIRR) has been used as a model of non-ICD inducer in the RNA-seq studies. However, this negative control has not been included in the validation studies. New experiments including this negative control should be performed, both in RT-PCR and flow cytometry experiments. Especially in flow cytometry analyses, this control is necessary to exclude a non-specific effect of the treatments, since the differences found in MCH-II protein expression (MFI) have been very modest.

Author Response

Comment #1. Apparently, the doses of ICD-inducers were the same for the lymphoma and breast cancer cell lines. In a previous study of the same laboratory the authors reported that RA/IFN-alpha induces ICD in lymphoma cell lines, but there are not available data about sensitivity the immunogenicity of this combination in MDA-MB-231 cells. Experiments demonstrating that RA/IFN-alpha induces ICD in MDA-MB-231 should be included.

Response #1: We thank the Reviewer for highlighting this important point. Currently, the Prof. Dolcetti’s lab is working on this topic and, in particular, we are collaborating with him to a study demonstrating the ability of RA/IFNα to induce immunogenic cell death (ICD) both in vitro and in vivo using several breast cancer cell lines (data not yet published). The relative manuscript is in preparation; therefore, herein we have performed new assays that indicate the potential of RA/IFNα to promote ICD in MDA-MB-231. We chose to analyze two among the DAMPs most frequently associated with ICD, the surface exposure of calreticulin (CRT) and heat-shock protein (Hsp70).

Please, find in the attached file the figures showing that 48 hours RA/IFNα treatment (as used in ref. Montico et al.) induced cell surface exposure of calreticulin (CRT) and Hsp70 in the breast cancer cell line included in the present study.

Comment #2. Gamma-irradiation (gIRR) has been used as a model of non-ICD inducer in the RNA-seq studies. However, this negative control has not been included in the validation studies. New experiments including this negative control should be performed, both in RT-PCR and flow cytometry experiments. Especially in flow cytometry analyses, this control is necessary to exclude a non-specific effect of the treatments since the differences found in MCH-II protein expression (MFI) have been very modest.

Response #2: We thank the Reviewer for this important comment. In Figure 3 Supplementary, we have now included new experimental data, obtained by both RT-PCR and flow cytometry analyses, that validate the lack of MHC Class II pathway down-regulation by gamma-irradiation in Mino cells.

Reviewer 2 Report

In this manuscript, Lamberti et al. study the association of miRNA and mRNA co-expression in Mino and MBA-MB-231 cells upon retinoic acid/IFNa and/or doxorubicin treatment to induce immunogenic cell death (ICD). Through their study, they identified and confirmed the down-regulation of MHC class II via miR-212-3p both at the gene and protein level and upregulation of MHC class I. Overall this work is interesting however I cannot recommend publication in biomedicines unless all the following comments are addressed:

  • In table 1, the authors present the up- and down-regulated miRNAs in both Mino and MDA-MB-231 cells. Why are there no miRNAs that are unique to each cell line?
  • In table 1, what does the NaN mean? Are the reported fold change values in log scale? And MDA-MB-231 is written as MDA-MD-231.
  • In Figures 1c and d, it is of crucial importance that the authors perform the RT-qPCR with γ irradiation as a control to be sure that the changes in miRNAs expression are due to induction of ICD. The authors should also include miR-222-5p in these two panels. Also, most of the reported significant changes from table 1 are not significant in Figures 1c and d, how do the authors explain this?
  • In table 2, are the reported fold change values in log scale? And MDA-MB-231 is written as MDA-MD-231.
  • The authors performed the initial experiments in both Mino and MDA-MB-231 cells. Why were none of the following experiments performed in MDA-MB-231 cells? But instead, these later experiments were performed with Mino and DOHH2 cells, why this change?   The authors should also perform the initial experiments with DOHH2 cells.
  • The authors should perform validation experiments showing the upregulation of MHC class I proteins.
  • In figure 6, the “a)” label is missing.
  • The legend of figure 7 had 2 descriptions for “b)” and none “d)”.
  • The experiments in figures 6 and 7 should contain the crucial γ irradiation control. 

Author Response

Comment #1. In table 1, the authors present the up- and down-regulated miRNAs in both Mino and MDA-MB-231 cells. Why are there no miRNAs that are unique to each cell line?

Response #1: In Table 1, we informed the Differentially Expressed miRNAs in Immunogenic Cell Death, which included those miRNAs whose expression was:

  • Significantly different in ICD inducers: DXR or RA/IFNα treated samples versus untreated ones (p ≤ 0.05, FC ≤ -1.3 for downregulated, FC ≥ 1.3 for upregulated).
  • Not significantly different in not ICD-inducer gamma-irradiation treated samples.
  • Regulated in the same way in both lymphoma and breast cancer cells.

Regarding those miRNAs that are unique to each cell line, this information could be found in the Upset plots presented in Figure 1. UpSet plots represent the intersections of a set as a matrix: each column corresponds to a set, and bar charts on top show the size of the set. For example, the first vertical column in Figure 1a indicates that there are 66 miRNAs downregulated “only” in mantle cell lymphoma cells subjected to RA/IFNα. Taking into consideration the huge number of possible intersections given that we were comparing 6 sets, we have removed the small intersections (<2 miRNAs in the intersection) and obtained plots that show us the main features of the data. If editors and reviewers consider it necessary, we could include supplementary tables showing miRNAs included in each intersection.

Please, find the table in the attached file.

Comment #2. In table 1, what does the NaN mean? Are the reported fold change values in log scale? And MDA-MB-231 is written as MDA-MD-231.

Response #2: In the table 1, we reported the Fold Change values only for the samples in which the p value < 0.05. Therefore, to make it clearer that it was, we change “NaN” (not available number) with NS (=not significant). However, if the Reviewer deems it necessary we can upload the table with all fold change values.

MDA-MB-231 is now well written.

Comment #3. In Figures 1c and d, it is of crucial importance that the authors perform the RT-qPCR with γ irradiation as a control to be sure that the changes in miRNAs expression are due to induction of ICD. The authors should also include miR-222-5p in these two panels.

Response #3. We thank the Reviewer for that relevant comment.

However, the short time allowed us for the revision (accordingly with Biomedicines regular revision time) and the delay in Taqman probes arrival (not yet available in our lab) prevented us from doing validations by qRT-PCR in γ-irradiated samples in a reasonable time. However, using the Taqman probes available in our lab, we have performed validation experiments for miR-4284, using U6 as normalization control.

The qRT-PCR results showed that although γ-irradiation induced a slight upregulation (1.387 ± 0.059) of miR-4284 in Mino cells, the increase was significantly lower than the upregulation induced by ICD-inducers (5.215 ± 0.065). Moreover, the miR-4284 expression did not exhibit a significant change in γ-irradiated DOHH2 compared to untreated cells (0.976 ± 0.020).

miR-4284 relative expression

Mino

DOHH2

γ-irradiation

1.387 ± 0.059

0.976 ± 0.020

RA/IFNα

5.215 ± 0.650

2.067 ± 0.270

If the Reviewer considers this information necessary for paper acceptance, we ask him/her to give us more time. However, no one among the upregulated miRNAs reported in table 1 was significantly modulated by γ-irradiation in our cancer models.

The potential involvement of miR-27a-5p and miR-222-5p in ICD-related MHC Class I up-regulation is currently under investigation in a separate study which involves a collaboration with Prof. Dolcetti’s lab. Indeed, Prof. Dolcetti’s group is characterizing in detail the modulation of antigen processing and presentation pathways by MHC-I during ICD. Moreover, previous reports have described MHC Class I modulation by ICD-based or non-ICD-based antitumor agents, e.g.: Zhou et al., 2021, PNAS (10.1073/pnas.2025840118), Takahashi et al., 2021, Front. Oncol (10.3389/fonc.2021.707473). On the other hand, in our initial real-time analyses we include the evaluation of miR-27a-5p whose decrement by ICD induction was previously described by Colangelo T. and collaborators (Colangelo T. Et al., Cell Death Dis. 2016 Feb 25;7(2):e2108. doi: 10.1038/cddis.2016.29. PMID: 26913599; PMCID: PMC4849155.)

However, herein we focused on the study of those ICD-upregulated miRNAs potentially implicated in the downregulation of MHC Class II, whose involvement in ICD has not been yet described and that could profoundly affect immune response.

Comment #4. Also, most of the reported significant changes from table 1 are not significant in Figures 1c and d, how do the authors explain this?

Response #4: As mentioned in a recent article (Coenye 2021, Biofilm, 10.1016/j.bioflm.2021.100043), there are several studies that have specifically addressed the correlation between results obtained with RNA-seq and qPCR. In particular, the study reported by Everaert et al, 2017, Sci Rep (10.1038/s41598-017-01617-3) demonstrated that 15–20% of genes are considered as “non-concordant” when results obtained with RNA-seq are compared to results obtained with qPCR (with “non-concordant” defined as both approaches yielding differential expression in opposing directions, or one of the methods showing differential expression while the other does not). However, of the genes showing non-concordant results, 93% show a fold change lower than 2 and ~80% show a fold change lower than 1.5. In addition, of the non-concordant genes with a fold change > 2, the vast majority are expressed at very low levels.

In this sense, here we noticed that those miRNAs whose expression was not concordant between RNA-seq and RT-qPCR assay exhibited a Cq>30 in the latter assays. Even when TaqMan assay was highly sensitive to detect those very low level of expression, it seems feasible that this fact could bring some concerns about reproducibility and bias. Anyway, we focus our research on the study of miR-4284 and miR-212-3p, whose expression and modulation were robust, consistent, and statistically significant. If the Reviewers and Editors of Biomedicines consider it necessary, we can expand discussion regarding this information in the manuscript.

Comment #5. In table 2, are the reported fold change values in log scale? And MDA-MB-231 is written as MDA-MD-231.

Response #5:

In table 2, the fold change values are in linear scale. The values of 0 <FC<1 were expressed as -1/FC.

If necessary we can upload the table with fold change in log scale.

MDA-MB-231 is now well written.

Comment #6. The authors performed the initial experiments in both Mino and MDA-MB-231 cells. Why were none of the following experiments performed in MDA-MB-231 cells? But instead, these later experiments were performed with Mino and DOHH2 cells, why this change? The authors should also perform the initial experiments with DOHH2 cells.

Response #6: We have formerly described a model of immunogenic apoptosis in which ICD was induced in lymphoma cell lines by the combination of retinoic acid (RA) and interferon-α (IFN-α) (Montico et al., 2017, Oncoimmunology, ref. 20). To investigate the potential involvement of miRNAs and their targets in mechanisms underlying the immunologic effects exerted by cancer cells subjected to ICD, here we initially performed RNA sequencing (RNA-Seq). We decided to include a breast cancer cell line to encompass a panel of diseases with different origin and phenotype, converging on miRNAs involved in both solid and hematological malignancies. In particular, we included the MDA-MB-231 cell line on the bases of a previous study by Sistigu et al. (Sistigu A et al., Nat Med. 2014 Nov;20(11):1301-9. doi: 10.1038/nm.3708. Epub 2014 Oct 26. PMID: 25344738) that described the ability of DXR to induce ICD in these cells. Overall, as addressed in Response #1, miRNA-seq data were integrated using bioinformatic analysis to identify Differentially Expressed miRNAs in Immunogenic Cell Death (Table I), which included those miRNAs whose expression was:

  • Significantly different in ICD inducers: DXR or RA/IFNα treated samples versus untreated ones (p ≤ 0.05, FC ≤ -1.3 for downregulated, FC ≥ 1.3 for upregulated).
  • Not significantly different in not ICD-inducer gamma-irradiation treated samples.
  • Regulated in the same way in both lymphoma and breast cancer cells.

Subsequently, inverse co-expression analysis was performed using RNA-seq data from RA/IFNα-treated Mino cells (Figure 3), to investigate potential miRNAs´ targets implicated in ICD subjected lymphoma cells that could be associated in the immunogenic behavior previously reported. Unfortunately, MDA-MB-231 cells don’t express MHC-II molecules, thus in the following experiments we can not use the breast cancer cell line. Therefore, given that RA/IFNα promotes ICD on DOHH2 lymphoma cell line (Montico et al., 2017, Oncoimmunology, ref. 20), we decided to include this model in the validation assays (Figure 6). As Reviewer suggested, we have now incorporated the analysis of miRNA modulation on this cell line as Figure 1 Supplementary. To note, miR7705 is not included in the Figure 1S because its expression was not detectable by real-time qPCR in DOHH2 cells.

Comment #7. The authors should perform validation experiments showing the upregulation of MHC class I proteins.

Response #7. As mentioned in Comment #3, the analysis of MHC Class I pathway during RA/IFNα- and DXR-induced is being extensively studied in the lab of Prof. Dolcetti and the relative results will be published in the next future. Moreover, some details about MHC Class I molecules modulation by ICD-based or non-ICD-based antitumor agents could be found in Zhou et al., 2021, PNAS (10.1073/pnas.2025840118) and Takahashi et al., 2021, Front. Oncol (10.3389/fonc.2021.707473). To address this statement, those references were added in the manuscript in Discussion section. On the other hand, we decided to focus on the regulation of MHC-II during ICD given that no information is available about this topic by the literature.

Moreover, as shown in Figure 5, in silico analysis of RNA-seq data obtained in samples treated with different ICD-inducers confirmed MHC-II downregulation in different cancer cell models. These findings induced us to focus our attention on investigation of ICD-mediated MHC-II modulation.

Comment #8. In figure 6, the “a)” label is missing. The legend of figure 7 had 2 descriptions for “b)” and none “d)”.

Response #8. We apologize for this mistake. Changes have been done.

Comment #9. The experiments in figures 6 and 7 should contain the crucial γ irradiation control.

Response #9. We thank the Reviewer for this important comment. In Figure 3 Supplementary, we have now included new experimental data that validate the lack of MHC Class II pathway modulation by gamma-irradiation in Mino cells.

Regarding the analysis of miR-4284 and miR-212-3p expression in γ-irradiated samples, we have answered in the Response to the Comment 3.

Round 2

Reviewer 2 Report

The authors have addressed all of my concerns with this revised manuscript. This manuscript can now be published in biomedicines.